# Non-Immune-Mediated, p27-Associated, Growth Inhibition of Glioblastoma by Class-II-Transactivator (CIITA)

**DOI:** 10.3390/cells13221883

**Published:** 2024-11-14

**Authors:** A Katherine Tan, Aurelie Henry, Nicolas Goffart, Christophe Poulet, Jacqueline A. Sluijs, Elly M. Hol, Vincent Bours, Pierre A. Robe

**Affiliations:** 1Department of Translational Neuroscience, University Medical Center Utrecht (UMCU) Brain Center, Utrecht University, 3584 CX Utrecht, The Netherlands; a.k.tan@umcutrecht.nl (A.K.T.); j.a.sluijs@umcutrecht.nl (J.A.S.); e.m.hol-2@umcutrecht.nl (E.M.H.); 2Department of Human Genetics, University of Liège, 4000 Liège, Belgium; a.henry@cergroupe.be (A.H.); goffart206@hotmail.com (N.G.); christophe.poulet@uliege.be (C.P.); vbours@uliege.be (V.B.); 3Laboratory of Rheumatology, University Hospital of Liège, University of Liège, 4000 Liège, Belgium; 4Department of Neurosurgery, University Medical Center Utrecht (UMCU) Brain Center, Utrecht University, 3584 CX Utrecht, The Netherlands

**Keywords:** glioblastoma, Class-II-Transactivator, proliferation, PI3-Akt, STAT3

## Abstract

Background: Previous works have shown that the expression of Class-II-Transactivator (CIITA) in tumor cells reduces the growth of glioblastoma (GB) in animal models, but immune effects cannot solely explain this. Here, we searched for immune-independent effects of CIITA on the proliferation of GB. Methods: Murine GL261 and human U87, GM2 and GM3 malignant glioma cells were transfected with CIITA. NSG (immunodeficient) and nude (athymic) mice were injected in the striatum with GL261-wildtype (-WT) and -CIITA, and tumor growth was assessed by immunohistology and luminescence reporter genes. Clonogenic, sphere-formation, and 3D Matrigel-based in vitro growth assays were performed to compare the growth of WT versus CIITA-expressing murine and human cells. Bulk RNA sequencing and RT^2^ qRT-PCR profiler arrays were performed on these four cell lines to assess RNA expression changes following CIITA transfection. Western blot analysis on several proliferation-associated proteins was performed. Results: The intracerebral growth of murine GL261-CIITA cells was drastically reduced both in immunodeficient and athymic mice. Tumor growth was reduced in vitro in three of the four cell types. RNA sequencing and RT^2^ profiler array experiments revealed a modulation of gene expression in the PI3-Akt, MAPK- and cell-cycle regulation pathways following CIITA overexpression. Western blot analysis showed an upregulation of p27 in the growth-inhibited cells following this treatment. PDGFR-beta was downregulated in all cells. We did not find consistent regulation of other proteins involved in GB proliferation. Conclusions: Proliferation is drastically reduced by CIITA in GB, both in vivo and in vitro, notably in association with p27-mediated inhibition of cell-cycle pathways.

## 1. Introduction

The infiltrative properties, substantial intratumoral heterogeneity, and conversion of the immune infiltrate into a tumor-supportive environment invariably lead to treatment resistance and short survival in IDH-wildtype glioblastoma, WHO grade IV (GB) [1,2,3,4,5]. The activation of T helper cells and the subsequent anticancer immune responses could be increased by strategies that enhance the presentation of potential tumor antigens in these aggressive brain tumors, such as presenting tumor-associated proteins on the cell surface by Major Histocompatibility Complex class II (MHC-II). Often, MHC-II expression in GB is low [6,7]. Class-II-Transactivator (CIITA) is a non-DNA-binding transcriptional coactivator that acts as the master regulator of MHC-II expression. It has been identified as a potential immunomodulatory target in anti-GB treatment, as GB cells that express CIITA show high expression of MHC-II proteins [6,7]. These MHC-II proteins harbor a rich repertoire of glioblastoma-associated antigens [8]. In line with this finding, CIITA overexpression in GL261 murine GB cells was reported to provoke tumor rejection in direct intracerebral xenograft experiments in syngeneic C57Bl/6 mice [9]. We, however, showed that these effects were not reproducible in adaptive cell transfer experiments and that CIITA overexpression in GB cells failed to significantly alter the activation of syngeneic, tumor-infiltrating immune cells in syngeneic human primary co-culture experiments [7]. These two findings suggested that CIITA could also hinder the progression of GB via other routes than activation of the immune system. CIITA has been reported to modulate key players of various non-immune genes and pathways involved in the proliferation of GB, such as NFKBIA or STAT3 in the NF-kappa B and STAT signaling pathways [10,11,12,13]. In this study, we thus investigated whether CIITA directly regulates GB proliferation.

## 2. Materials and Methods

### 2.1. Glioblastoma Cell Cultures and Transfection

Four cell lines expressing CIITA were used: GL261, a murine GB cell line, and three human cell lines, U87 (American Type Culture Collection [ATCC] accession number HTB14), GM2, and GM3. The generation of these latter early-passage human glioma cell lines, GM2 and GM3, has been described previously [14,15]. Cell characteristics were confirmed by SNP array and key mutation analysis. Stable CIITA overexpression was established using a CIITA-encoding pcDNA3.1 plasmid (-CIITA), as described previously [7]. Empty pcDNA3.1 was used as a control (-wildtype, -WT). Cells were cultured in a humidified incubator at 37 °C and 5% CO_2_ atmosphere in DMEM/F-12 medium, supplemented with 10% fetal calf serum (FCS) under constant selection pressure using 0.5 mg/mL G418 antibiotics (all Gibco™ (ThermoFisher Scientific, Waltham, MA, USA)). Cell surface expression of MHC-I and -II following CIITA transfection was checked with flow cytometry analysis using the anti-H-2 K/D class I monoclonal antibody (clone M1/42, BioLegend, San Diego, CA, USA) and the purified anti-mouse I-A/I-E antibody (ImTec diagnostics, Antwerp, Belgium) and isotype-matched antibodies as negative control. Flow cytometry was performed in the BD FACSAria™ II Cell Sorter (BD Biosciences, San Jose, CA, USA).

GL261 cells transduced with a luciferase plasmid were used for a bioluminescence assay. Briefly, viral supernatants of transfer lentiviral plasmid pLenti6-IRES-Luc were used for transduction, and transduced cells were selected with 1 mg/mL Blasticidin (Sigma^®^, St. Louis, MO, USA). They were injected intracranially as described previously [16].

### 2.2. Mice

We used five- to seven-week-old immunodeficient athymic nude mice, Crl:NU-*Foxn1^nu^*, and NOD/SCID gamma (NSG) immunodeficient mice obtained from Charles River^®^ animal facilities (Charles River Laboratories^®^, Wilmington, UK). The mice were housed in sterilized filter-topped cages at the animal facility of the University Hospital of Liège. All animals were cared for in compliance with the guidelines of the Belgium Ministry of Agriculture and the ethical committee’s laboratory animal care and use regulation (Directive 2010/63/EU of the European Parliament and of the Council of 22 September 2010 on the protection of animals used for scientific purposes), following approval by the animal experiment ethics committee of the University of Liege. For mice tumor growth experiments, GL261-wildtype and GL261-CIITA cells were used (see below).

### 2.3. Cell Count and Clonogenic Assay

Cells from all four cell lines (-WT and -CIITA) were seeded in a 6-well plate at 75,000 cells per well in DMEM/F-12 medium with 2% FCS. At day 7, cells were trypsinized and counted on a hemocytometer using trypan blue dead cell exclusion (Gibco™). For the clonogenic assay, 2000 cells were plated in a T75 cell culture flask. After seven days, colonies were counted in the densest area of the flask with 4× magnification on an AxioVert A1 inverted microscope (Zeiss™, Oberkochen, Germany).

### 2.4. Sphere Formation Assay

GL261-WT and GL261-CIITA (2 clones) were cultured at low density in a low adherence plate and in serum-free medium to stimulate sphere growth (DMEM/F-12, supplemented with 1% penicillin/streptomycyin (Gibco™)) and after seven days, sphere size was measured for 4 to 6 spheres per condition. The experiments were performed in triplicate.

### 2.5. Three-Dimensional Glioma-Matrigel^®^-Based Cultures

GL261, U87, GM2, and GM3 cells (-WT and -CIITA) were counted and resuspended as 10,000 cells/μL in medium (DMEM/F-12 supplemented with 10% FCS and 1% penicillin/streptomycin). Then, 15 μL of Matrigel (Corning^®^ Matrigel^®^ Growth Factor Reduced- Corning Life Sciences bv, Amsterdam, The Netherlands) was mixed with 5 μL of cell suspension in the medium, and droplets were placed in the middle of a (non-adherent) 48-well plate. The droplet was incubated at 37 °C with 5% CO_2_ for 60 min when the medium was added. Preparations were kept in culture for up to 8 days. Overall cell viability was measured using the CellTiter-Glo^®^ Luminescent Cell Viability Assay (Promega, Madison, WI, USA), according to the manufacturer’s instructions. Values were expressed in relative luminescence units (RLUs).

### 2.6. In Situ GB Xenograft Model Using Immunodeficient NSG and Nude Mice

Fifty thousand GL261-wildtype or GL261-CIITA cells suspended in 2 μL phosphate-buffered saline (PBS, Gibco™, Grand Island, NY, USA) were injected in the right striatum of NSG (3 mice per group) and nude mice (5 mice per group) using a stereotactic frame (coordinates with respect to the bregma: ML −2.5, AP −1, DV −3) as described previously [15].

Bioluminescence imaging with a luciferase reporter was used to follow tumor growth in the NSG mice. Briefly, D-Luciferin 150 mg/kg was injected intraperitoneally during induction with isoflurane 2.5% (in 100% oxygen). For imaging, mice were placed in the prone position and were imaged every 7 days starting at day 21, using a charge-coupled device camera-based system for bioluminescence imaging (Xenogen IVIS-50 system; exposure times 10–30 s, binning 8, field of view 12, f/stop 1, open filter, Xenogen, Alameda, CA, USA). Analysis was performed by manually determining the region of interest around the head and measuring it as photons/second/square centimeter/steradian (p/s/cm^2^/sr) after 15 s of exposure time for both groups. Images were processed using Living Image ^®^ (IVIS imaging systems, Xenogen) and Igor Pro^®^ Software (version 2.5, WaveMetrics^®^, Lake Oswego, OR, USA).

Nude (athymic) mice were sacrificed 21 days after intrastriatal GL261 implantation, and the resulting tumor volume in their brains was assessed by histology as described previously [17].

We did not perform survival experiments, as this was deemed unethical by the Ethical Committee of the University of Liege. In line with the University of Liege recommendations, mice were held alive until the planned end-date of the experiments, unless they had to be euthanized due to tumor-related unmanageable signs of discomfort.

### 2.7. Western Blot Analysis and HGF ELISA

One million sub-confluent cells per line were used for protein extraction. Cell pellets were lysed with 1x SDS loading buffer (consisting of 2× Loading buffer [100 mM Tris pH 6.8, 4% sodium dodecyl sulphate [SDS], 20% glycerol] and diluted 1:1 with suspension buffer [0.1 M NaCl, 0.01 M Tris-HCl with pH7, 6 and 0.001 M EDTA dissolved in MilliQ]) in a 95 °C heating block for 5 min. After the cells were lysed, the DNA was sheared with a 25 G needle. Proteins were loaded and run on SDS-polyacrylamide-based gels (SDS-PAGE, #1610301 and #1610158, Bio-Rad, Hercules, CA, USA), using percentages adapted to the target protein molecular weight. After electrophoresis, proteins were transferred onto a nitrocellulose membrane (GE Healthcare, Chicago, IL, USA) using wet blotting in a tank transfer system (Bio-Rad). Membranes were incubated overnight at 4 °C with primary antibodies in a gelatin-based blocking buffer (for 500 mL, MilliQ-based: 3.03 g, 4.5 g NaCl, 1.25 g gelatin, 2.5 mL Triton X-100, pH 7.4). The primary antibodies used were CIITA (7-1H Santa Cruz Biotechnology, Dallas, TX, USA), STAT3 (Cell signaling #9139), pSTAT3 (rabbit, Cell signaling #9145), ERK1/2 (Cell signaling #9106), pERK1/2 (Cell signaling #9106), p27 (BD Biosciences), Phospho-NF-kB p65 (Ser536) (Cell signaling #3031), NF-kB p65 (Cell signaling #8242), MEK1/2 Antibody (Cell signaling #9122), phospho-MEK1/2 (Cell signaling #9121) and GAPDH (loading control, mouse Sigma-Aldrich). The secondary antibodies used were mouse or rabbit IR800 (LI-COR biotechnology, Lincoln, NE, USA) or AF647 (Jackson ImmunoResearch, West Grove, PA, USA). Fluorescence was assessed using the Odyssey^®^ scanner (LI-COR Biosciences, Lincoln, NE, USA). We quantified the Western blots in FIJI [18], measuring pixels and intensity of the bands for the protein of interest and GAPDH. Then, we divided the protein of interest by the GAPDH values to correct for variations in loaded protein. We used the ELISA human HGF assay following manufacturers’ instructions (KAC2211, Invitrogen, Waltham, MA, USA).

### 2.8. RNA Sequencing Data and RT^2^-Profiler™ Array

For RNA sequencing, RT^2^-profiler™ array experiments, cell pellets from U87, GM2, GM3, and GL261 cells (-WT or -CIITA) were collected at 80% confluency from T75 flasks and processed for RNA extraction using the QIAzol lysis reagent (Qiagen, Hilden, Germany), followed by chloroform precipitation (Merck, Sigma-Aldrich, St. Louis, MO, USA). The Agilent 2100 bioanalyzer (Agilent Technologies, Palo Alto, CA, USA) was employed with the Agilent RNA 6000 Nano Kit (Agilent Technologies) to analyze the integrity of the RNA. A total of 1000 ng RNA per sample was used as input. Library preparation was performed with the Illumina Truseq^®^ mRNA HT kit (Illumina Inc., San Diego, CA, USA). Purified libraries were quantified by qPCR with KAPA Library Quantification Kits (Kapa Biosystems, Wilmington, MA, USA). Sequencing was performed using Illumina’s NextSeq 500 system, running for 75 cycles. We analyzed genes that were differentially expressed (log2 fold change >1 or <−1 at an adjusted *p*-value < 0.01) using the KEGG 2021 database [19,20,21] and biological processes from the Gene Ontology database [22,23] through Enrichr^®^ (Ma’ayan lab, center for bioinformatics, Icahn School of Medicine at Mount Sinai, [24,25,26]). RT^2^ Profiler™ PCR Arrays for panels ‘Human Cell cycle’ and ‘Human Cell Death PathwayFinder’ were performed following the manufacturer’s instructions (RT^2^ Profiler™ PCR Array, Qiagen Benelux B.V., Antwerp, Belgium).

### 2.9. Statistical Analyses and Data Visualization

Most of the data were analyzed with Microsoft Excel. Statistical analysis was performed using R-Studio (RStudio Team (2020). RStudio: Integrated Development for R. RStudio, PBC, Boston, MA, USA) and GraphPad Prism version 10.0.0 for Windows, (GraphPad Software, Boston, MA, USA), The Mann–Whitney test was applied for non-parametric data, while the non-paired Student’s *t*-test was used for normal distributed data. One-way Analysis of Variance (ANOVA) was used for multiple comparisons. Bar graphs and boxplots represent the mean with separate datapoints and/or ±SD. RNA sequencing data were analyzed using the DESeq2 method (DESeq2 R package v1.25.9, [27]) to obtain the differential expression statistics between the controls and each cell type. RStudio was used for data visualization. Heatmaps represent the log2fold change from the differentially expressed genes, which were regulated in the same way in all cell lines depicted in the heatmaps. Final figures were created using Adobe Illustrator (Adobe^®^, Mountain View, CA, USA).

## 3. Results

### 3.1. CIITA Expression in GL261 Xenografts Drastically Reduces Tumor Growth in Immunodeficient Mice

First, we injected luciferase-expressing GL261-WT(-Luc) or luciferase-expressing-GL261-CIITA(-Luc) cells in the striatum of fully immunodeficient NSG mice (Figure 1A for set-up) and followed tumor growth by luminescence measurement every week starting at day 21. GL261-WT-Luc cells formed large tumors (average radiance 2833 ± 1418 p/s/cm^2^/sr, Figure 1B) that justified the euthanasia of the bearing mice after 21 days, and prior to the next scheduled measurement of day 28. In contrast, there was a considerable delay in tumor growth when GL261-CIITA-Luc cells were injected (5.5 ± 3 p/s/cm^2^/sr, *p* = 0.07, Figure 1B). It indeed took 61 days for GL261-CIITA-injected mice to develop tumors large enough to produce significant signs of discomfort. To avoid missing data due to potential mouse death, it was decided to perform the final bioluminescence measurement on day 61 instead of the planned 63rd day, and to euthanize the mice without further delay. At that moment, bioluminescence measurement showed that the tumors had grown to volumes similar to those of WT-injected mice at day 21 [3433 ± 757 p/s/cm^2^/sr, *p* = 0.56, Figure 1B).

Intrastriatal tumor graft experiments were then performed on (athymic) nude mice using GL261-WT or GL261-CIITA cells (Figure 1A for set-up). At 21 days, five of five GL261-WT-injected mice had developed large, macroscopically visible tumors, while only microscopically visible tumor engraftment was seen in the mice that were injected with GL261-CIITA cells (*p* = 0.0079, Χ^2^-test, Figure 1C and [App cells-13-01883]).

### 3.2. CIITA Inhibits Glioma Cell Proliferation in 2D and in 3D In Vitro Cultures

GL261-CIITA cells repeatedly grew 50.12 ± 18.8% less profusely after seven days in monolayer culture than GL261-WT cells, as assessed by cell counts with trypan blue exclusion tests (*p* = 0.06, n = 4, Figure 2A). Likewise, in 2D clonogenic assays, only 81 ± 8.8 colonies formed after seven days in GL261-CIITA versus 218.75 ± 74.2 colonies in GL261-wildtype cultures (*p* = 0.029, n = 4, Figure 2B). In tumor sphere formation assays, CIITA-spheres (of two separate clones) were also reproducibly smaller and unevenly formed after 7 days, as compared to wildtype spheres (GL261-CIITA-C1 33.1 ± 6.6 µm and GL261-CIITA-C2 35.4 ± 9.2 µm versus GL261-WT 71.2 ± 15.2 µm) (*p* < 0.0001, one-way ANOVA, Figure 2C,D). Finally, GL261-CIITA grew significantly slower in 3D conditions in Matrigel than their wildtype equivalents at day 8 (53,748 ± 42,611 RLU versus 252,942 ± 65,768 RLU, *p* = 0.016, Figure 2F). Similar results were observed in human malignant glioma cultures. Proliferation was consistently decreased in two of three tested human GB cell lines, with a 1.5-fold reduction in cell counts after seven days for U87-CIITA and a 1.4-fold reduction for GM3-CIITA compared to their respective wildtype controls (*p* = 0.015 and *p* = 0.02, respectively, Figure 2E). In 3D Matrigel cultures, the 3D Matrigel-embedded structures of U87-CIITA and GM3-CIITA also grew less than their wildtype controls (355,949 ± 11,959 [U87-CIITA] versus 732,830 ± 187,665 [U87-WT], *p* = 0.03, and 841,741 ± 48,910 [GM3-CIITA] versus 1,219,948 ± 264,696 [GM3-WT], *p* = 0.07, Figure 2F). However, we did not observe any reproducible effects on the proliferation of CIITA overexpression in GM2 cells.

### 3.3. RNA Sequencing Analysis Shows Downregulated Genes Involved in Pathways Known in GB Proliferation

All cell lines showed CIITA and HLA-II or MHC-II protein expression following CIITA transfection (tested by Western blot, as shown in Figure 3C and [App cells-13-01883], and FACS analysis, as shown in [App cells-13-01883]).

At the RNA level, as assessed by RNA sequencing, the murine GL261 cells differentially expressed 12 genes of the MHC-I and -II family following CIITA transfection ([App cells-13-01883]). Likewise, in human GB cells, 17 HLA-I and -II genes were differentially expressed in U87 and GM2 following CIITA transfection, and 3 HLA-I and -II genes in GM3 following this treatment ([App cells-13-01883]).

Other CIITA-induced, non-MHC-related changes in RNA expression were found in the four tested GB cell lines (the GL261 murine glioma cell line and three human cell lines, U87, GM2, and GM3). These differentially expressed genes were further analyzed using Enrichr^®^ [24,25,26]. For all sequenced cell lines individually, biological processes and pathways involving the antigen-processing machinery were commonly upregulated in all CIITA-expressing cells. Pathways involved in proliferation, such as the Phosphatidylinositol 3-Kinase (PI3)-Akt and the MAPK(inase) pathways, were likewise downregulated in all cell types ([App cells-13-01883] with results from KEGG 2021 pathway analysis and biological process Gene Ontology). Of note, these pathway alterations resulted from the up- and downregulation of different sets of genes in the different cell types tested (GL261, U87, GM2, and GM3).

Four genes were similarly upregulated in all lines following CIITA transfection: TMEM184A, PLAC1, ENDOD1, and NFATC2 (Figure 3A). One gene, MAP7D2, was differentially expressed in all cell lines, but was not consistently up- or downregulated in all cell lines. In the human GB cells U87, GM2, and GM3, 51 genes were differentially expressed in all three cell lines, of which 38 genes were regulated similarly: 14 downregulated and 24 upregulated (Figure 3A,B). Pathways involved in cellular senescence were associated with upregulated genes, and downregulated genes were associated with pathways involved in proliferation ([App cells-13-01883]). For example, Hepatocyte Growth Factor (HGF), the ligand of the tyrosine kinase c-MET receptor, was downregulated in all three human cell lines. 

We further assessed the effect of CIITA transfection on the expression of more specific gene panels using RT^2^ profiler arrays. On the ‘Human Cell cycle’ panel array, both caspase (CASP)3 and CDK6 were downregulated in all four cell lines. On the ‘Human Cell Death Pathway Finder’ array, BCL2L11 was upregulated, and caspases (CASP)1, −3, −7 were likewise differentially downregulated in all cell lines following CIITA transfection. Of note, Skp2 and CDK8 were unchanged and upregulated in GM2, respectively, but were downregulated in the other cell lines ([App cells-13-01883]).

### 3.4. Pathway Alterations Following CIITA Transfection

Following our cell culture experiments and our RNA-sequencing-based observation that CIITA regulates pathways involved in proliferation, we analyzed the modulation of several receptor tyrosine kinases, their downstream effectors, and several cell-cycle inhibitors using Western blots and ELISA (Figure 3C,D and [App cells-13-01883]). PDGFR-beta was downregulated in all cell lines (Figure 3C,D), while c-Met, MEK1/2, and phosphorylated (p)MEK1/2 remained unchanged in the human GB cells ([App cells-13-01883]). We likewise did not find any reproducible evidence that ERK phosphorylation was modulated by CIITA transfection ([App cells-13-01883]). STAT3 and pSTAT3 were variably modulated by CIITA expression in the different cell types ([App cells-13-01883]). CIITA did not modulate I-kappaB-alpha (NFKBIA) protein expression in our cell lines ([App cells-13-01883]). We could not identify any amount of HGF in our cell lines using either Western blot or ELISA tests.

Finally, the cell-cycle inhibitor p27 was consistently upregulated in U87, GM3, and GL261 cell lines but not in GM2 (Figure 3C,D).

## 4. Discussion

CIITA expression has been reported to induce malignant glioma tumor rejection by immune-related mechanisms [9,28]. We, however, challenged this paradigm by showing that these results were not reproducible in cell transfer experiments and that CIITA-expressing human GB cells did not elicit any strong activation of tumor-infiltrating lymphocytes in human syngeneic co-cultures [7]. Here, we show that the intracerebral growth of murine GB xenografts expressing CIITA is also strongly delayed in nude mice (lacking only T lymphocytes) and in NSG mice (lacking all immune cell types). Although we did not use survival as an outcome—as this was deemed unethical by the ethical committee of the University of Liège as tumor growth (especially in GL261-WT) could lead to major impairments and suffering in the mice—it is clear in both mice experiments that tumor growth is delayed. Moreover, in NSG mice, the timeline in Figure 1B shows significant tumor growth delay in the mice implanted with CIITA-expressing GL261 cells with respect to those implanted with WT-GL261 cells. This confirms that immune mechanisms are not solely responsible for CIITA-mediated GB rejection.

Our in vitro growth experiments in 2D, spheroid assays, and 3D-Matrigel^®^ preparations reveal that CIITA can act as a potent inhibitor of proliferation in GB. While to the best of our knowledge, a direct effect of CIITA on cell proliferation has not yet been reported in the literature, ChIP-seq experiments have shown that CIITA modulates several genes besides those of the MHC complex in leucocytes, including proliferation-related genes like STAT3 and NFKBIA [13]. While we did not find evidence of the modulation of these two genes in our RNA sequencing experiments, four non-MHC genes were constantly upregulated in all four tested GB cell lines following CIITA transfection: TMEM184A, PLAC1, ENDOD1, and NFATC2. Of those, PLAC1 and NFATC are unlikely to mediate the observed antiproliferative effects of CIITA in GB. Indeed, PLAC1 was reported to enhance the proliferation of cervical and nasopharyngeal cancer, and PLAC1 is associated here with poor prognosis [29,30] and can promote invasion in breast cancer and nasopharyngeal carcinoma [30,31]. Likewise, NFATC2, a nuclear factor, is generally overexpressed in GB and is associated with GB proliferation, survival, migration, and invasion [32]. On the contrary, however, TMEM184A, a heparin transmembrane receptor, shows antiproliferative properties through PDGF inhibition and inactivation of the ERK pathway [33,34]. ENDOD1 was also reported as a candidate tumor suppressor and inhibits proliferation, migration, and invasion in prostate cancer [35]. This gene was not reported earlier in GB research and may deserve further exploration beyond the scope of this work. When looking at human cells only, several additional genes were modulated in common in response to CIITA transfection. Of these, HGF, the ligand of the tyrosine receptor c-MET, has been associated with GB proliferation (reviewed in [36]) and is a major determinant of poor prognosis [37]. We, however, did not measure HGF at the protein level in our GB cells by Western blot and ELISA assays, and the expression of its receptor c-MET was not regulated by CIITA, which makes it unlikely that this growth factor plays a significant role in the antiproliferative role of CIITA in practice.

Targeted qRT-PCR (RT^2^) gene expression panels showed additional alterations in the expression of cell-cycle regulating genes following CIITA transfection in GB cells, including the downregulation of CDK6 and CASP3. CDK6 is a fundamental regulator of the cell-cycle G1-S transition and a major determinant of GB proliferation [38]. Caspase (CASP)3 was recently shown to present non-apoptosis-related functions and to enhance cell proliferation (reviewed in [39]). Next to these gene-specific common alterations, at the pathway level, both our RNA sequencing and RT^2^-profiler arrays experiments showed CIITA-induced alterations in the expression of genes involved in pathways that regulate the proliferation via the PI3-K and MAPK pathways, senescence, and cell survival in GB cells.

At the protein level, based on the reported effect of TMEM184A on PDGFR signaling [33], and the known importance of this growth factor receptor in glioma proliferation [40,41], we assessed the expression of PDGFR-beta. Western blots showed a reproducible downregulation of this receptor in all tested cell lines following CIITA overexpression. However, downstream this receptor, the modulation of MEK/ERK or pSTAT3 signaling alterations following CIITA transfection was neither consistent nor correlated with the antiproliferative effect of this treatment.

On the contrary, the protein p27^KIP^ was upregulated in the three GB cell types (GL261, U87, and GM3) that were growth-inhibited following CIITA transfection. p27^KIP^ is a major inhibitor of the cell-cycle progression and inhibits several Cyclin-dependent kinases (CDKs). p27^KIP^ is controlled via the ubiquitin-ligase Skp2, which targets p27^KIP^ for proteolytic degradation (reviewed in [42]), a cascade that is facilitated by CDK8 [43]. This sequence could help explain why we did not observe the antiproliferative effects of CIITA in GM2, in contrast to U87-, GM3- and GL261-CIITA cells. In our RT^2^ profiler array, both Skp2 and CDK8 were either unchanged or slightly increased in GM2 cells while consistently downregulated in the other three cell lines. Thus, this asymmetrical modulation of p27, Skp2 and CDK8 by CIITA overexpression between GL261, GM3 and U87 cells on the one hand, and GM2 cells on the other hand could contribute to its anti-proliferative effects in the former, and lack thereof in the latter. Such differences in response to treatments between GB cell lines are well known, making generalized treatment for these tumors especially difficult. Further work is thus warranted to define the potential of Skp2 and CDK8 as markers of sensibility for the personalized treatment of GB using CIITA.

Our results have an implication for further laboratory work and perhaps in the future for clinical trials. In preclinical experiments, the effects of CIITA on tumor growth should further be assessed prior to concluding on its potential for antitumor immune vaccination strategies. However, if this technique finds its way to the clinic, the distinct anti-proliferative role of CIITA may prove to be particularly relevant, as CIITA restoration could thus also act in highly immunocompromised patients (typically, those treated symptomatically with immunosuppressive doses of corticosteroids). However, further research is first needed to assess the potential value of potential resistance markers to CIITA-mediated growth inhibition (e.g., SKP2) to select or stratify the patients in future clinical trials.

## 5. Conclusions

We previously reported that the immune-stimulatory effects of CIITA in GBM were at best limited and likely did not solely explain the rejection of CIITA-expressing xenografts in immunocompetent mice [7]. As we here show that CIITA-expressing xenografts also grow very poorly in severely immunodeficient mice, it becomes obvious that CIITA effects in GB do not solely rely on the immune system. Our results show that CIITA transfection can increase p27 expression and directly impairs GB cell proliferation. These results can provide guidance for the design of future trials that will assess the restoration of CIITA for the treatment of glioblastoma.

## Figures and Tables

**Figure 1 cells-13-01883-f001:**
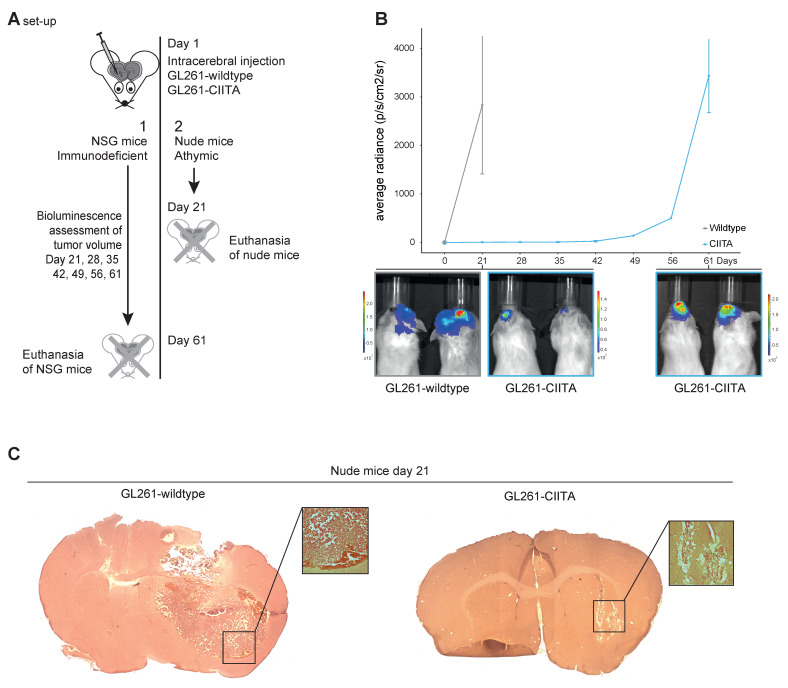
(**A**) Experimental setup of in vivo intracerebral xenograft experiments with NSG (fully immunodeficient) and nude (athymic) mice. (**B**) Luciferase imaging in NSG (immunodeficient mice), n = 3 per group. Growth kinetics showing average radiance (in photons/second/square centimeter /steradian [p/s/cm^2^/sr]). Wildtype (grey line) injected NSG mice grow large tumors within 21 days, while CIITA (blue line) injected NSG mice show a large delay in tumor growth, until day 42, when the tumors start to grow exponentially. Bioluminescent images of the mice are used to show tumor growth of -wildtype and -CIITA at day 21 and for -CIITA at day 61. (**C**) Brains of nude mice showing large tumors after injection with GL261-WT and only microscopic tumor engraftment after injection with GL261-CIITA with microscopic detail of a region of interest.

**Figure 2 cells-13-01883-f002:**
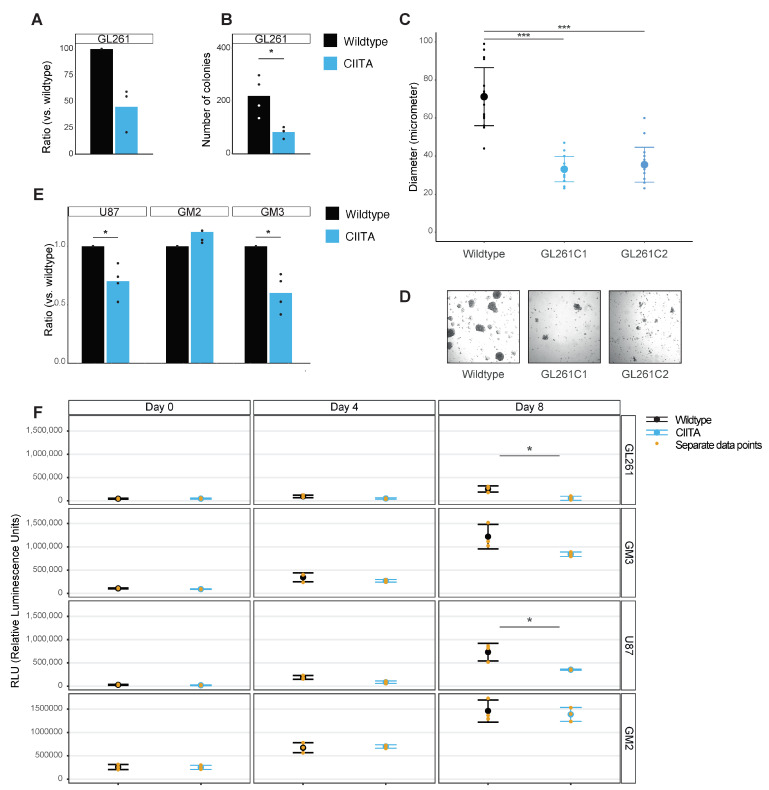
(**A**) Cell count assay of GL261 (murine) glioma cells in monoculture, Y-axis as ratio of GL261-CIITA versus GL261-WT. (**B**) Colony formation assay in GL261 cells, Y-axis depicting the number of colonies per condition (WT versus CIITA). All bargraphs represents mean with separate data points, * *p* < 0.05. (**C**) Boxplots representing mean ±SD and sepa-rate datapoints as size of individual spheres formed in a sphere formation assay, in grey GL261-WT cells, and in two tones of blue two clones (C1 and C2) of GL261-CIITA. *** *p* < 0.0001. (**D**) Representative images of the spheres from GL261-WT, -CIITA clone 1 and -CIITA clone 2. (**E**) Similar cell count assay as (**A**), but in human glioblastoma cell lines: U87, GM2 and GM3 * *p* < 0.05. (**F**) Boxplots representing mean ± SD with values representing relative luminescence units (RLU) * *p* < 0.05.

**Figure 3 cells-13-01883-f003:**
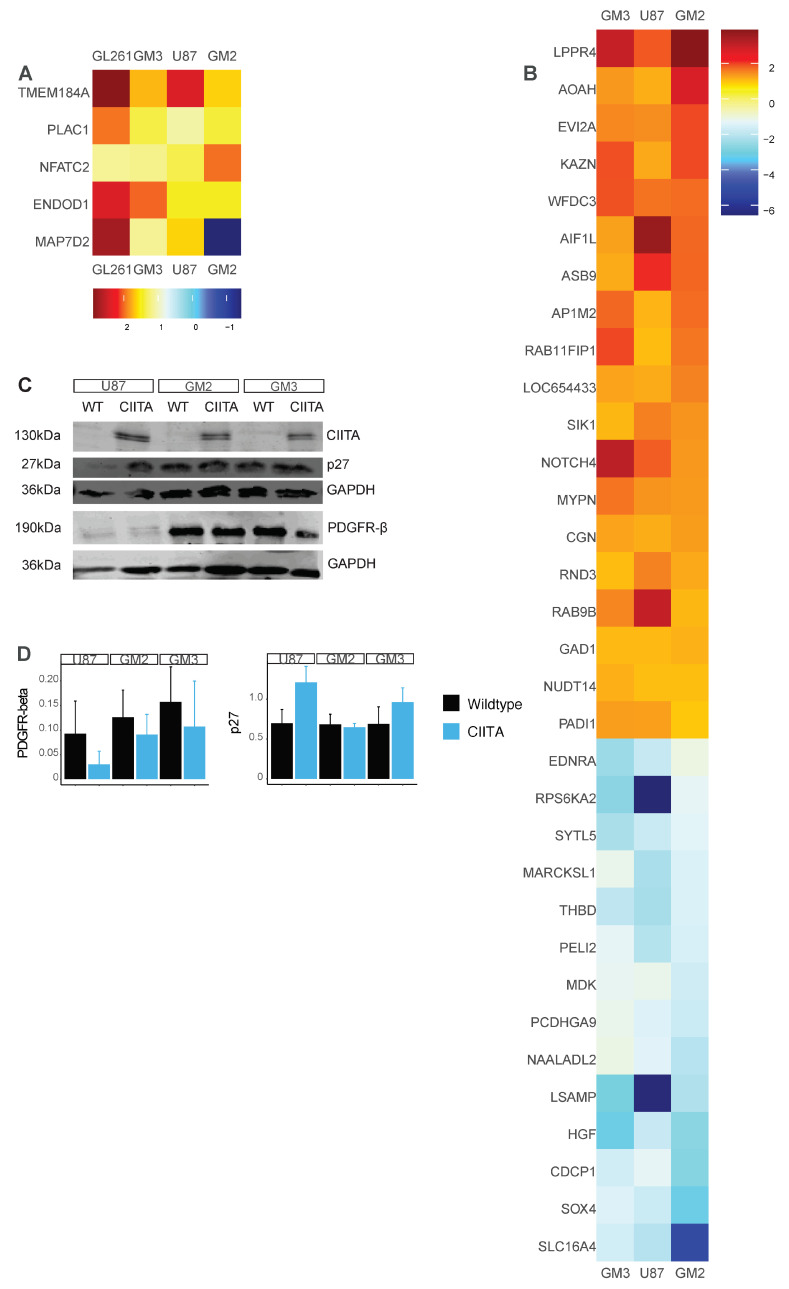
(**A**) Heatmap showing five differentially regulated genes in the four tested glioma cell lines: murine GL261, human U87, GM3 and GM2. Four of those genes were consistently upregulated in all four cell lines. (**B**) Heatmap showing all genes differentially up- and downregulated in the human cell lines: U87, GM2 and GM3. Note that genes which are differentially expressed in all four cell lines are not listed here (but in (**A**)), and HLA-F, an MHC-related gene is showed in [App cells-13-01883]. (**C**) Western blot experiments, showing protein expression in human cell lines, differentiating between -WT and -CIITA overexpression. pSTAT3 = phosphorylated STAT3 (**D**) Bargraphs representing mean ± SD of protein expression (of three separate western blot experiments).

## Data Availability

All original data are readily available upon request by contacting the corresponding author.

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
