# Peer review of "Non-Immune-Mediated, p27-Associated, Growth Inhibition of Glioblastoma by Class-II-Transactivator (CIITA)"

_cells, 2024, doi:10.3390/cells13221883_

Round 1
Reviewer 1 Report
Comments and Suggestions for Authors
Major comments:
In this study the authors tried to establish the immune-independent effects of Class-II-Transactivator (CIITA) on the proliferation and growth of glioblastoma multiforme (GBM) tumors both in vitro and in vivo.
1. In the Introduction authors should include a statement to introduce CIITA and its role on MHC- Clas-II genes. For example " CIITA is a non-DNA-binding transcriptional coactivator, that plays a crucial role in regulating the expression of Major Histocompatibility Complex (MHC) class II genes."
2. In the line 41, the authors should write IDH mutant glioblastoma multiforme (GBM) and not "gliomas" They have used incorrect nomenclature. All gliomas are not GBM as low grade-gliomas are not glioblastoma multiforme (GBM). Please change mention of all "GB" to "GBM" throughout the entire manuscript maintaining the conventional name.
3. In Figure 1C, the authors should include histology of brains of remaining 2 mice engrafted with either GL261-WT and GL261-CITA cells.
4. In legends of Fig 1A, it is incorrect to mention the experiment as luciferase assay. The authors should change the legend as "Luciferase imaging of Gl261 intracranially implanted NSG mice.
5. The authors should explain why was a survival analysis for mouse experiments not done in the results.
6. The authors should use better H&E stained images for Figure 1C. The current image is not acceptable and lacks clarity.
7. Authors should consider including more mice (at least n= 5) in Figure 1 experiment to conclude that CIITA expression in GL261 xenografts drastically reduces tumor growth in immunodeficient mice. Just using 2 mice for the data is not enough.
8. The authors clam there was a considerable delay in tumor growth when GL261-CIITA cells were injected ( Figure 1A-1B). The claim "It took three months for GL261-CIITA-injected mice to develop tumors of volumes that were comparable to those of WT-injected mice at day 21 is not supported by any bioluminescence images. Authors should include appropriate images to support this claim. It appears from Figure 1B that median average radiance in GL261-CIITA-injected mice in 61 days is more that wild-type GL261 injected mice imaged in 21 days. This contradicts the author claims.
9. In figure 2D are those 3 separate images taken on same day or different days? No information regarding the time point of these images are provided in figure legends. Authors should include the timepoints.
10. Why would the CIITA overexpression in GM2 cells offer no significant anti-proliferative effect of GM2 cells? Is that connected to the absence of downstream immune-regulatory machinery in GM2 cells? Authors should provide an explanation for this observation in the study. Can they conclude something from the RNA-seq data?
11. STAT3 is frequently found to be constitutively active (phosphorylated) in many types of cancers, promoting tumor growth, survival, and proliferation. In suppl Fig S2A, western blot data, the authors observed CIITA overexpression in GM2 significantly reduce p-STAT3. Why reduction of p-STAT3 did not affect proliferation of GM2 cells?
12. The authors reasoned that higher basal expression of SKP2, in GM2 cells could explain the lack of p27KIP mediated anti-proliferative effects of CIITA overexpression in these cells since SKP2 targets p27KIP for proteolytic degradation. In suppl figure S2D (RT2 Profiler PCR Array results), what is the statistical level of significance (p value) between the difference of SKP2 in U87 and GM2 cells? The authors should include statistics in Fig S2D to justify their claim.
13. There is no comment on how this CIITA overexpression in GBM cells can be translated in patients in clinical trials who already presents with lethal GBM tumors. A few sentences should be included.
Minor comments:
In the abstract in the line write "Tumor growth was reduced in vitro. Currently "in vitro" is not in italics. In the abstract and text convert all glioblastoma (GB) to glioblastoma (GBM)
Author Response
Please see the attachment.
We would like to alert that – although saved in high resolution, the quality of the images that are added to the Word file, is lower than the quality of the separately saved PDF files for the images. Therefore, we would like to ask the reviewers to, when assessing the figures, look at the separate files for Figures and Supplementary Figures.

Reviewer 2 Report
Comments and Suggestions for Authors
-
- The main question of this paper is whether the expression of Class-II-Transactivator (CIITA) in tumor cells which reduces the growth of glioblastoma in animal models is dependent on an intact immune system.
- CIITA has been shown to inhibit brain tumor growth in animal models, but the mechanism of action is unclear. In this paper the applications of CIITA were studied in mice either immunodeficient or athmyic nude mice, and the results demonstrate that CIITA is active in these animals in the inhibition of glioma growth. Yet CIITA has been shown to stimulate a potent anti-tumor immune response. In this paper CIITA which presumably works by stimulating CD4+ cells is still effective in immunodeficient and athymic mice, which suggests that another mechanism of action may be taking place independent of the immune system.
- This paper is suggestive of alternative antitumor actions of CIITA independent of the immune system.
- More work needs to be done to determine the antitumor actions of CIITA.
- CIITA does seem to work well as an antitumor agent in animal models, but it is not used in the clinical setting for unclear reasons.
Minor editing of the English would be reasonable.
Author Response

(The authors gave the same response as above.)

Reviewer 3 Report
Comments and Suggestions for Authors
The paper entitled "Non-immune-mediated, p27-associated, growth-inhibition of glioblastoma by Class-II-Transactivator" examines the impact of CIITA on glioblastoma proliferation. The authors found that CIITA overexpression significantly reduced intracerebral growth of murine GBM cells in immunodeficient mice, as well as in vitro growth of 3 cell lines. Mechanistically, CIITA upregulated the cell cycle inhibitor p27 and also affected the PI3-Akt, MAPK, and cell cycle pathways.
To further strengthen this work, the authors should consider the following:
-
Inclusion of a survival curve in Figure 1 to provide a more comprehensive assessment of the in vivo findings.
-
Conducting a rescue experiment using siRNA against p27 to directly demonstrate its role in the observed growth inhibition.
-
Performing a more in-depth analysis of the PI3K and MAPK pathways using western blots to elucidate the phosphorylation status of key signaling molecules.
-
Expanding the amount of data presented, as the current dataset is quite limited.
Overall, this study provides novel insights into the growth-inhibitory effects of CIITA in GBM, and the suggested experiments would help solidify the mechanistic understanding.
Comments on the Quality of English LanguageOK
Author Response

(The authors gave the same response as above.)

Round 2
Reviewer 1 Report
Comments and Suggestions for Authors
Majority of my comments and concerns have now been addressed by the authors. However, one major point below need to be addressed.
In Fig 1A, the authors should include BLI images of all 3 NSG mice (since n= 3). This is especially important since n=3 is a low number of mice for any experiment. It is acceptable for a grant not sufficient for a research manuscript.
Also what made the authors decide not to include the BLI imaging of nude mice data? Is BLI done only for NSG mice? If yes, why didn't the author didn't do imaging for nude mice? Why the nude mice were all terminated in 21 days?
How long the therapy increased survival in them. The outcome measures of shrunken tumor growth with BLI is incomplete without the 3rd mouse. Also the way the outcome was measured in NSG mice (with BLI) was different from nude mice (histology was done). This is a shortcoming of the methodology unless a clear explanation is not provided. The authors should include mouse brain histology all nude mice in the experiment (n =5). Figure 1C currently is incomplete as they inlcuded only one pair of brain sections from control (GL-261 WT) and GL-261-CIITA group. Histology date of 1 mouse each group showing treatment response is not enough for publication.
In Figure 1D use “Euthanize nude mice” and Euthanize NSG mice” instead of using “kill” There is a difference between meaning of the words “kill” and “euthanize”
In support of their response authors should provide evidence ( a letter stating survival experiments following treatment in mouse model are not allowed as a policy) from their protocol approved by the animal experiment ethics committee of the University of Liege. The authors should acknowledge that survival experiments are vital components to assess efficacy of glioblastoma treatment.
There is no clear explanation why animal experiment ethics committee protocol would not allow survival experiments in GBM mouse model following a novel treatment. Such experiments are permitted and performed in institutes all over the world including United States and Europe.
